# Changes in Community-Dwelling Elderly’s Activity and Participation Affecting Depression during COVID-19 Pandemic: A Cross-Sectional Study

**DOI:** 10.3390/ijerph20054228

**Published:** 2023-02-27

**Authors:** Miki Tanikaga, Jun-ichi Uemura, Fumiko Hori, Tomomi Hamada, Masahiro Tanaka

**Affiliations:** 1Department of Occupational Therapy, College of Life and Health Sciences, Chubu University, 1200 Matsumoto-cho, Kasugai 487-8501, Aichi, Japan; 2Department of Integrated Health Sciences, Graduate School of Medicine, Nagoya University, 1-1-20, Daiko-minami, Higashi-ku, Nagoya 461-8673, Japan; 3Department of Nursing, College of Life and Health Sciences, Chubu University, 1200 Matsumoto-cho, Kasugai 487-8501, Aichi, Japan; 4College of Business Administration and Information Science, Chubu University, 1200 Matsumoto-cho, Kasugai 487-8501, Aichi, Japan; 5Department of Rehabilitation, Faculty of Health Science, Nihon Fukushi University, 26-2 Higashihaemi-cho, Handa 475-0012, Aichi, Japan

**Keywords:** activity, participation, depression, COVID-19, community-based rehabilitation, community-dwelling elderly

## Abstract

We determined the changes in the activity or participation of the community-dwelling elderly in Japan during the COVID-19 pandemic and identified the activities leading to depression. This will allow us to evaluate rehabilitation interventions that can be used to minimize or eliminate the negative impact of COVID-19 on today’s community-dwelling elderly. Herein, demographics, activity or participation (Activity Card Sort-Japan version: ACS-JPN), the number of social networks (Lubben Social Network Scale: LSNS), and depression (Geriatric Depression Scale: GDS) were examined in 74 community-dwelling elderly in Japan from August to October 2020. A statistical analysis was conducted to determine the effect of demographics on GDS, LSNS, and ACS-JPN, to compare the activity retention rates of the four domains using ACS-JPN, and to extract the activities that might affect depression using a generalized linear model. The results show that the retention of leisure activity with a high physical demand (H-leisure) and sociocultural activities was significantly lower than instrumental activities of daily living and leisure activity with a low physical demand (L-leisure). L-leisure and the number of social networks were possible risk factors for depression during the pandemic. This study highlighted the importance of maintaining the number of L-leisure and social networks at home to prevent depression in community-dwelling elderly when they could not perform outdoor activities and direct interpersonal interaction.

## 1. Introduction

The coronavirus disease (COVID-19) pandemic forced people worldwide to refrain from going outdoors and partake in outdoor activities to prevent the spread of infection. As a result, the elderly in the community were particularly deprived of their important outdoor activity and social interactions in the wake of COVID-19 [1]. In Japan, the infection rate began to soar by the end of March 2020, with an alarming increase in severe infections. Eventually, the government declared a state of emergency in April 2020, soliciting voluntary restraint from unnecessary activities and interpersonal exchanges [2]. Although the activity restrictions were voluntary in Japan, based on individual judgment and not strong curbs by the government, such as a lockdown, the physical activity time of community-dwelling elderly strongly decreased [3].

Meanwhile, depression in the elderly increased during the COVID-19 pandemic [4]. A study investigating the relationship between physical activity and depression in the elderly during the COVID-19 pandemic reported that elderly individuals who were more physically active had lower levels of depression [5]. Similarly, a rapid systematic review suggested that community-dwelling elderly who frequently engaged in multiple physical activities during the COVID-19 pandemic were less likely to experience depressive symptoms [6]. Numerous studies have examined the relationship between activity or participation and depression among the elderly. They concluded that factors inversely associated with the degree of depression include the duration, intensity, and level of physical activity [7,8], participation in various leisure activities [9], different health-promoting behaviors [10], and the frequency of and variation in social activities [11]. In addition, some studies have described that a higher satisfaction with participation in leisure and other activities was associated with lower rates of depression [9,12].

However, previous studies examined the association between depression and activities such as games, social activities, cultural activities, outings, communication, lifestyle, volunteering, and hobbies individually [9,10,11]. An examination of few activities may confound the effects of other activities. For example, a review on mental health during the COVID-19 pandemic has stated that lifestyle changes have influenced mental health deterioration [13], but the scope of activity domains examined was limited to sleeping and eating habits. Community-dwelling elderly usually engage in multidisciplinary activities in their daily life, including the instrumental activities of daily living, leisure activities with low or high physical demands, and social interactions. Therefore, it is important to reveal what domain of activity or participation affects their mental health to make interventions that consider their actual situation.

Activity Card Sort (ACS) can be used to assess the activity and participation quantitatively. Since physical activity intensity is associated with depression [7], this study used the activity items included in the Activity Card Sort-Japan version (ACS-JPN) [14] for the assessment of leisure activities by distinguishing them into those with high- and low-physical loads in order to have a detailed understanding of the impact of the COVID-19 pandemic on the elderly who were forced to restrict their activities. The ACS was primarily developed in the United States and has since been modified worldwide based on cultural and social contexts [15,16,17,18,19]. The ACS-Australia was also used as an evaluation index to examine the relationship between physical function and activity or participation in the community-dwelling elderly [20].

Furthermore, behavioral restrictions to prevent COVID-19 infection caused social isolation among community-dwelling elderly [21]. As social isolation is associated with a poorer mental health and is one of the risk factors for depression [22,23,24], undesirable physical and mental health effects are significant problems faced by the community during the COVID-19 pandemic [4,25]. The previous study has indicated that objective social isolation includes a lack of social networks and social inactivity [26]. Similarly, previous studies before the COVID-19 pandemic reported that a lack of social networks and social inactivity, indicators of social isolation, were associated with the quantity and quality of activity or participation among community-dwelling elderly [27,28], and voluntary work, cultural activities, hobbies, and shopping successfully predict social connectedness among elderly [29]. However, the avoidance of direct interpersonal relationships and the digitization of interpersonal relationships in the current epidemic situation may change (or is changing) the nature of both social networks and social activities. Therefore, social isolation should be divided into two concepts: a lack of social networks and social inactivity. Moreover, their relationship to depression should be examined. Therefore, this study included the extent of social networks (Lubben Social Network Scale) and the extent of social participation (ACS-JPN) as independent variables.

This study aimed to understand the changes in the activities and participation patterns of community-dwelling elderly during the COVID-19 pandemic and to identify the activity domains that might lead to depression.

## 2. Materials and Methods

### 2.1. Study Design

A cross-sectional study was conducted using a questionnaire survey from August to October 2020 during the COVID-19 pandemic, and the collected data were analyzed using statistical methods. We followed the Strengthening the Reporting of Observational Studies in Epidemiology (STROBE) guidelines for conducting and reporting the results of this cross-sectional observational study [30].

### 2.2. Participants

We included 107 community-dwelling older adults (both men and women) aged at least 65 years who belonged to the senior citizen’s club in Ishiodai, Kasugai City, Aichi Prefecture, Japan. Ishiodai is a Japanese suburban area with an aging population of 46.7% [31]. A senior citizen’s club is a community-based organization where senior citizens gather voluntarily to engage in activities. Such clubs are run on membership fees and receive support from the national and local governments. These clubs thrive to enrich the elderly’s lives by offering them a “purpose of life” and health through companionship and leisure activities. Group activities such as monthly birthday parties, karaoke, handicrafts, “Go”, and “tai chi” are performed in the local meeting place.

The questionnaires were distributed to all club members, and response to the survey forms was considered as consent for this study. Although we briefed the club officers on our research, it was impossible to screen out potential participants since the personal information of the club members was unavailable. Therefore, a questionnaire with instructions was distributed to all members, and the participants were selected based on the eligible criteria.

This study excluded the elderly individuals who were hospitalized or admitted within the past year, were employed on a full-time basis, had been certified for long-term care as per the long-term care insurance, had been diagnosed with dementia, or had a physical injury or disease that interfered with daily life activities (e.g., neurodegenerative disease and organic brain disorder).

The primary author had been practicing intergenerational exchange in this senior citizens’ club for the past five years and was acquainted with the officers and members of the club. Therefore, the response rate was ensured by asking the club members to distribute and collect the questionnaires.

### 2.3. Procedure and Outcome Measures

From August 2020 to October 2020, the club officers distributed envelopes containing the survey forms to all club members, and collection boxes were set up in the community meeting places. The envelopes contained the questionnaire, a letter explaining the purpose and nature of the study, the possibility of publishing the results, the option to voluntarily withdraw from answering the questions, and a return envelope. The letter and questionnaire clearly stated that responding to these questions would be considered as consent for this study.

The ACS-JPN was used to investigate the changes in activities and participation among community-dwelling elderly due to COVID-19. The ACS was initially developed in the United States to evaluate the implementation of a wide range of activities among adults [32] and has since been used as an important assessment measure for community-dwelling elderly [33,34,35]. The ACS has already been used in over ten countries, including Australia, the United Kingdom, and Hong Kong, considering different cultural and historical contexts. The reliability and validity of ACS and its versions have been thoroughly examined. Cronbach’s alpha coefficients for the four ACS domains show a high internal consistency of instrumental activities of daily living (IADLs): 0.93; leisure activity with low-physical demand (L-leisure): 0.92, leisure activity with high physical demand (H-leisure): 0.94, and sociocultural activity: 0.83 [34]. The intraclass correlation coefficient (ICC) for retest reliability was 0.79 for all domain activities score; domain-wise, IADLs had the highest ICC (0.89), whereas the H-leisure domain had the lowest (0.71) [34].

The ACS-JPN comprised 72 items of occupation, including 26 IADLs, 18 L-leisure, 10 H-leisure, and 18 sociocultural activities; the number of current and previous activities can be quantitatively evaluated in each of these four domains [14]. Furthermore, the activities included in the ACS-JPN were selected from the everyday activities of community-dwelling elderly in five diverse regions of Japan, allowing for sufficient content validity of the scale [14].

Participants were asked to select from the following five alternatives about each activity included in the ACS-JPN compared to the year before the COVID-19 pandemic: “I do the same”, “I do less”, “I have stopped doing”, “I have newly started”, and “I have never done before”. Then, as per Baum et al.’s manual, the rate of changes in the activities performed before and during the COVID-19 pandemic (activity retention rate) was calculated by dividing the activity performance score at the time of the survey for each occupation by the score collected before the pandemic [32].

The activity implementation status score at the time of the survey was calculated as 1 point for “I do the same”, 0.5 points for “I do less”, 0 points for “I have stopped doing”, and 1 point for “I have newly started” [32]. Activities that have never been performed before were excluded. Moreover, as for the implementation status score for the year before the pandemic, activities for which the subject answered, “I do the same”, “I do less”, or “I have stopped doing” were regarded as implemented before the pandemic and were scored as 1. For the scores calculated this way, the activity retention rate was calculated by dividing the implementation status score at the time of the survey, excluding “I have newly started” activities, by the pre-pandemic implementation status score [32]. In other words, the maximum value of the activity retention rate was 1. For each type of activity and participation status, ACS-JPN determined the rate of activity continuation compared to the implementation status at the time of the survey and before the pandemic.

The Lubben Social Network Scale-Japanese short version (LSNS-6) was used to assess the number of social networks. The LSNS-6 is a 6-item 6-point Likert scale used to examine the size and frequency of the respondent’s interpersonal interaction networks. A score of ≤12 points indicates that the person is socially isolated [36,37]. The LSNS-6 has a Cronbach’s alpha of 0.82, a correlation coefficient for repeatability, *r* = 0.92, and excellent interrater reliability (ICC = 0.96, 95% confidence interval, CI: 0.90–0.99) [37]. Recent studies have frequently used the LSNS-6 to examine the degree of social isolation of community-dwelling elderly in Japan [38,39].

Next, the 15-item Geriatric Depression Scale-Japanese version (GDS-15-J) was used to investigate the level of depression. The GDS-15-J comprises 15 questions to be answered as “yes” (score = 1) or “no” (score = 0); the higher the total score (range: 0–15), the greater the level of depression [40]. The GDS-15-J has a cutoff point of 6/7 when used as a clinical screening test for depression [41]. The scale has high internal consistency (Cronbach’s alpha: 0.83), a significant item-total correlation (*p* < 0.001), and >0.30 for all items, except item 15. It is frequently used to examine depression levels among community-dwelling older adults in Japan; for example, it has been used in cross-sectional studies of the relationship between mindfulness and depression among community-dwelling older adults in Japan and studies of factors affecting social frailty [42,43].

The sociodemographic characteristics of the participants were collected, which included age, sex, living arrangements (living alone, living with a spouse, living with unmarried children, etc.), usage status of care insurance, employment status, history of hospitalization within the past year, previous medical record, and whether they receive assistance from others to perform their daily activities. Additionally, to ascertain the level of independence in everyday life and leisure activities and suitability to the study criteria, we used the Japan Science and Technology Agency Index of Competence (JST-IC). The JST-IC is a self-administered questionnaire developed in response to recent lifestyle changes in community-dwelling elderly due to technological advancements and measures their ability to relate to this modern life [44,45]. It comprises 16 items to be answered as “yes” or “no”, with one point given for “yes”: the higher the total score, the higher the activity capacity. The reliability and validity of JST-IC have been examined (Cronbach’s alpha = 0.86), and standardized values have been developed for ages 65–84 years based on data obtained from 2573 elderly individuals in Japan [45,46]. Furthermore, in a large Japanese longitudinal study of community-dwelling older adults, the JST-IC was used as an index of activity capacity among this population to analyze the relationship between activity capacity and cognitive frailty and to extract factors that influence participation in meaningful activities [47,48].

### 2.4. Data Analysis

All statistical analyses were carried out using the Statistical Package for the Social Sciences (SPSS) for Windows (version 25.0, IBM Corp., Armonk, NY, USA). In addition, continuous variables of the eligible participants were tested for normal distribution using the Shapiro–Wilk test.

We conducted a group comparison to confirm whether attributes such as age, sex, and living arrangements affect depression, the number of social networks, and activity retention rate before and after the pandemic. Pearson’s correlation, unpaired t-test, and one-way analysis of variance were used for normally distributed data. In contrast, Spearman’s correlation, Mann–Whitney U, and Kruskal–Wallis tests were used for non-normally distributed data. In addition, the Friedman test was used to compare the activity retention rate for the four ACS-JPN domains compared with pre-pandemic levels, and Wilcoxon’s signed-rank test with Bonferroni correction was used for the post hoc test.

Next, factors affecting depression during the pandemic were examined using Poisson regression analysis with a generalized linear model (GLM), with the GDS-15-J as the dependent variable, the LSNS-6 and ACS-JPN activity retention (four domains) as independent variables, and age, sex, and living arrangement as the adjusted variables. Poisson regression was selected because the GDS-15 data had a non-normal Poisson distribution. Poisson regression can simultaneously handle categorical and continuous variables as explanatory variables. The crude model revealed the ways in which the focal factors of this study (IADL, L-Leisure, H-Leisure, sociocultural activity, and LSNS-6) affected the participant’s depression levels without covariates of other demographical factors. Further, models 1, 2, and 3 adjusted for age, sex, and living arrangement were tested and compared with the crude model to determine whether these focal factors affected the depression levels via correlation or cross classification with respect to each demographical factor. Model 1 was adjusted for age, Model 2 was adjusted for age and sex, and Model 3 was adjusted for age, sex, and living arrangement. There are some studies reporting the association between the characteristics of the older adults (age, sex, and living arrangement) and their activity and participation [1,49]. Missing data for dependent, independent, and adjustment variables were excluded from the sample. A *p*-value of <0.05 was considered as statistically significant.

### 2.5. Ethical Considerations

This study was approved by the Ethical Review Committee of Chubu University (Approval number: 20190097-3). We verbally explained the purpose and content of this study to all officers of the senior citizen clubs. Each member was enclosed with an explanatory note and a questionnaire, and it was clearly stated that the answer to the questionnaire would be regarded as consent to participate in this study. The study instructions also indicated that participation in this study was voluntary and that no one would be disadvantaged if they did not participate; that they could withdraw from this study at any time; that they could not delete the data after the analysis was completed; that all data would be handled in encrypted form; that the questionnaire would take approximately 20 min to complete; and that they would not be forced to answer the questionnaire if it was physically or mentally demanding.

## 3. Results

### 3.1. Participant Characteristics and Activity Implementation Status

Of 107 community-dwelling older adults from the senior citizens club, 81 responded to the questionnaire (response rate: 75.7%). Four of these respondents required nursing care, and three had missing values in the responses. Therefore, 74 (69.2%) people were included in the final analysis (mean age: 78.6 (SD = 5.70) years, 40 males, 34 females). None of the respondents were working full-time.

A total of 20 people lived alone, of which, 15 (75%) were females, 36 lived with a spouse only, and 18 with other family units. The mean JST-IC score was 11.4 (SD = 3.17), which was higher than the national mean score of 9.7 (SD = 4.2) (difference = 1.7 points), indicating that our study sample had activity capacities that exceeded the national average for daily living capacities of community-dwelling elderly [46].

Using the 6/7 cutoff value for the GDS-15-J, 12 participants (16.2%) were depressed [41]. Meanwhile, the LSNS-6 cutoff value (11/12) showed that 16 participants (21.6%) were in the socially isolated category [36]. Moreover, 15% of the elderly living alone and 24.1% of the elderly living with a family were socially isolated.

The mean activity and participation scores as measured by the ACS-JPN at the time of the survey were: IADLs (26 items): 14.90 (SD = 4.85), L-leisure (18 items): 8.64 (SD = 3.13), H-leisure (10 items): 3.67 (SD = 1.89), and sociocultural activity (18 items): 7.20 (SD = 3.06). During the pandemic, five respondents (6.8%) did not perform H-leisure activities, and one (1.4%) did not engage in any sociocultural activity. All data collected from the 74 respondents are summarized in Table 1.

Age, LSNS-6, and implementation of L-leisure, H-leisure, and sociocultural activity were usually distributed. However, data distribution had non-normality in GDS-15-J, IADL implementation, and the four domains of activity retention calculated by ACS-JPN. A sex-related difference was noted only in the status of IADL implementation at the time of the survey. Among the 26 IADLs in the ACS-JPN, men showed a mean of 13.6 (SD = 4.85), and women showed a mean of 16.5 (SD = 4.56); the difference was significant (*p* = 0.012). Accordingly, Table 1 shows the results by sex for the status of IADL implementation.

There was no statistically significant relationship between these demographic factors and the activity retention rate calculated by ACS-JPN, the level of social isolation by LSNS-6, or the level of depression evaluated using the GDS-15-J. The results of the analysis of the effects of different demographics such as age, sex, and living arrangement on the GDS-15, LSNS-6, and ACS-JPN are listed in Table 2.

### 3.2. Changes in the Number of Activities Compared to Pre-Pandemic Status

The mean activity retention rates at the time of the survey compared with those before the COVID-19 pandemic were 0.86 (SD = 0.24) points for IADL, 0.82 (SD = 0.27) points for L-leisure, 0.67 (SD = 0.32) points for H-leisure, and 0.64 (SD = 0.34) points for sociocultural activities; in all activity domains, decreases occurred during the pandemic, according to the pre-pandemic reference value of 1. Items with a ≥30% decrease in activity retention were “shopping for clothes/shoes” and “using public transport” in IADLs, “taking photographs”, “handcrafting/knitting”, “coloring books/collage of pieces of colored paper”, and “karaoke” in L-leisure, and “taking a day trip”, “going to a garden/park”, “going to a spa”, and “traveling” in H-leisure. The maximum number of activities affected was in the sociocultural domain: “visiting family/friends who are ill”, “visiting friends/acquaintances”, “eating out”, “doing activities with children/grandchildren”, “going to places of worship”, “gathering with family/relatives”, “going to a coffee shop with family or friends”, “attending a community/senior group”, “going to alumni”, “studying for personal advancement”, “volunteering”, and “going to the library” (Appendix A).

In the activity domains, there was a statistically significant difference in the activity retention rates, with H-leisure and sociocultural activities being significantly lower than IADL and L-leisure (*p* < 0.001) (Figure 1).

### 3.3. Effect of Activity Retention and the Number of Social Networks on Depression in the COVID-19 Pandemic

The Poisson regression analysis using a GLM showed that activity retention rates of L-leisure and the number of social networks were significant, influential factors for depression during the pandemic (Table 3). The results were similar with and without the age, sex, and living arrangement as adjustment variables.

## 4. Discussion

### 4.1. Participant Characteristics and Activity Implementation Status

Globally, the prevalence of depression among community-dwelling elderly has been reported to be 17.1%, while 28.1% of the community-dwelling Japanese elderly have depression [50,51]. At the same time, the prevalence of depression among our study participants was relatively low, at 16.2%. In previous studies measuring social isolation among community-dwelling elderly using the LSNS-6, a 21.2% prevalence was reported in Germany and 49.8% in Malaysia, 31.0% of elderly living alone and 24.1% of those living with family in previous Japanese studies [52,53,54]. However, in this study, only 15.0% of the elderly living alone and 24.1% of the elderly living with their family were judged as socially isolated. A possible explanation for this low prevalence of depression and social isolation among the elderly who lived alone in this study is that the participants belonged to a senior citizens club, which comprises a group of people who participate in activities and interpersonal relationships, allowing them to interact with other members of the society outside of their family.

Furthermore, the results of the JST-IC activity capacity analysis showed that the mean score was 11.4 (SD = 3.17), which was higher than the national mean score of 9.7 (SD = 4.2) [46], indicating that our study sample had a higher activity capacity than the standard Japanese community-dwelling elderly. It is known that socially active people have a lower risk of developing depression, which was further corroborated by our results [55]. However, it is noteworthy that the statistics for the prevalence of depression and social isolation were based on the studies conducted before the COVID-19 pandemic, and their results may not be completely comparable. The levels of depression and social isolation in society have changed drastically compared to the pre-pandemic era [56]; however, since ours is a cross-sectional study during a COVID-19 pandemic, the extent of change remains unexplored.

Furthermore, a sex-related difference was observed only in IADLs measured using the ACS-JPN, and the number of IADLs in women was higher than in men. However, no sex-related differences were observed in L-leisure, H-leisure, or sociocultural activities. Moreover, age and living arrangements did not affect any of the four domains of ACS-JPN. Previous reports suggest that women traditionally tend to perform household chores as part of their role at home [57], which may explain the women’s participation in IADLs, even among the elderly. In addition, IADLs is one of the activities necessary for maintaining community life among the domains of “activities and participation.” Women formed a more significant proportion of the elderly living alone, which might influence this result.

### 4.2. Comparison of Activity Retention Rates before and during the Pandemic

Compared to the pre-pandemic status, we found a considerable decrease in activity retention rates in all domains of the ACS-JPN. After the Japanese government declared a state of emergency in April 2020, all activities of the senior citizens’ clubs were suspended. At the time of this study, some activities had resumed with restrictions, such as limiting the number of participants. However, curbs on activities with a high infection risk, such as eating, drinking, and karaoke, were still in place. The H-leisure and sociocultural activities of the ACS-JPN included several activities that were performed outside the home and involved interpersonal interaction [14]. During the survey, the government continued to call for self-restraint in activities; accordingly, the community-dwelling elderly avoided “going out” as a leisure activity and social interaction with others, which lowered the overall activity retention rate in all four domains.

Furthermore, comparing the four activity domains, the activity retention rate was significantly lower for H-leisure and sociocultural activities than for IADLs and L-leisure. Since it was not the purpose of this study, the reasons for the decrease in outdoor activities and activities involving interaction with others could not be found in this study. However, the decline in the retention rate for these domains was greatly influenced by the environmental impact of infection prevention measures put up during the pandemic. This could be due to requests from the local government for infection prevention [2] or the fear of infection [58]. The correlation between the frequency and anxiety of going out among community-dwelling elderly detected in a survey conducted during the COVID-19 pandemic [1] supports this consideration.

This study quantified the activity retention rates using the ACS-JPN to classify and measure activity or participation in four domains to understand the specific changes in activity or participation patterns during the pandemic.

In addition, the more affected activities in the four ACS-JPN domains included IADLs of “shopping for clothes” and “using public transportation” and the L-leisure, “karaoke”, both of which were deemed unnecessary outings and activities by the Japanese government associated with a high risk of infection, probably decreasing the activity retention rate. Notably, “karaoke” was eliminated due to the associated high infection risk, even though it is commonly performed by the Japanese elderly, resulting in lower retention rates for L-leisure [14].

### 4.3. Factors Associating Depression in the Community-Dwelling Elderly during the COVID-19 Pandemic

The GLM results show that the activity retention rate of L-leisure activities and social isolation as the number of social networks were associated with depression during the pandemic. Among the four domains, only L-leisure activities were significantly associated with depression during the COVID-19 pandemic, suggesting the importance of low physically demanding leisure activities at home for mental health. A study investigating leisure activities and well-being during the pandemic suggested that maintaining time for leisure activities that were easy to perform during the pandemic can help to maintain a sense of general well-being, which concurs with our study results [59].

In addition, the number of social networks was a factor influencing depression. This factor is a variable (LSNS-6) determined by the frequency of interpersonal interactions and the number of partners. The effects of the degree of social isolation on depression are supported by the existing literature [60]. Social isolation comprises two factors: the number of social activities and social networks [26]. In this study, social activity was not associated with depression, but the social network was. The reason why the number of social networks was associated with depression was that, although electronic media such as telephones and videophones were frequently used as interaction tools during the COVID-19 pandemic [61], there were individual differences in the social interactions that community-dwelling elderly had with family, relatives, and friends using these electronic media [62]. Those who could use electronic media may have maintained a greater number of social networks than those who could not, despite restrictions on going outside and direct interpersonal interactions. There was a strong sense of anxiety in Japan about infection due to the lack of information on COVID-19 and the lack of prevention and treatment for COVID-19 (at the time of the survey). Therefore, social networks shared the sense of anxiety that both sides had and may have had a buffering effect on these anxieties. It is possible that social networks led to a decrease in anxiety, which, in turn, led to the prevention of depression.

In this study, neither H-leisure and sociocultural activities nor IADL retention rates were associated with depression. Opportunities for social participation and physically demanding activities are often reported to prevent depression since they involve a change in scenery outdoors and interaction with others [55,63,64]. On the other hand, an IADL decline has also been reported to correlate significantly with depression [65]. These differences may be attributed to the cross-sectional nature of this study during the COVID-19 pandemic; when the highly infectious virus was widespread, there was no obvious treatment, and the elderly at a high risk of serious illness refrained from acting anyway [66], which was significantly different from the previous studies. In this study, all participants uniformly stopped performing H-leisure and sociocultural activities to prevent infections, so this study did not identify the association with depression.

In addition, the type and content of H-leisure and sociocultural activities conducted, as well as the preparedness of participants, may have changed during the pandemic. A qualitative study during the COVID-19 pandemic revealed that community-dwelling elderly avoided interpersonal interactions for fear of infection and had negative psychological conditions such as anxiety [67]. In addition, community-dwelling elderly took a new perspective on activities and participation, such as re-engaging in creative leisure activities [68], rediscovering family time and bonds [69], and innovating communication and using technology [70]. Therefore, sociocultural activities, a variable that affected depression before the COVID-19 pandemic, were not found to be an influencing factor in this study, not only because of a consistent and significant decrease in the number but possibly also because of qualitative changes. Furthermore, the participant was originally a group of physically and mentally functional people who could perform IADLs well. Even with COVID-19, they needed to perform them for their lives and had a high activity retention rate. Hence, it was challenging to find an association with depression.

In this way, it is important for the quality health promotion of community-dwelling elderly to discuss the relationship between activity participation, social isolation, and the degree of depression in light of such an infectious pandemic. Further prospective longitudinal studies are warranted to consider the long-term impact of the COVID-19 pandemic.

### 4.4. Practical Implications and Future Research

In this section, we discuss the assessment and practical interventions by rehabilitation professionals to prevent a decline in L-leisure activities and social isolation during such pandemics for reducing the risk of depression based on existing evidence and the results of this study.

Recently, the concept of health promotion has gained popularity, emphasizing that true health is not merely the absence of a disease but involves the promotion of a better state of being and the provision of support to the elderly in living a life with purpose [71]. However, despite the spread of this concept of health, more efforts from the geriatric rehabilitation professionals are needed to analyze the relationship among activity or participation, the number of social networks, and depression levels in the community-dwelling elderly, especially now that their activity has been restricted to prevent the spread of COVID-19 infection and social isolation has increased [21]. By incorporating the unique perspective obtained by performing a comprehensive activity assessment and activity domain-specific examination, this study examined the relationship between activity restraint and mental health during the COVID-19 pandemic. Furthermore, we aimed to provide essential information on rehabilitation intervention based on the activity or participation (occupational therapy) to minimize or eliminate the negative impact of COVID-19 on the community-dwelling elderly. Therefore, our result will help the rehabilitation professionals in selecting appropriate intervention methods during and after the COVID-19 pandemic.

The results of this study emphasize the importance of maintaining low-physical-load activities and the number of social networks to prevent depression when individuals, regardless of sex, are forced to refrain from their daily outdoor and interpersonal interactions. Monitoring participation in physical and leisure activities has been reported to be a predictor of mental health status and is thus essential for depression prevention [8,72]. In addition, those who exercise more frequently were more strongly associated with depression than those who carry out only housework [73]. When external factors limit outdoor activities during a disaster such as COVID-19, it is important to prevent depression by maintaining, at a minimum, low-physical-load activity and personal interaction with others via video calls or letters. Fortunately, this has been possible recently using digital platforms, phones, and the Internet [74]. The effectiveness of technology to allow interpersonal interaction without physical contact has been realized during the pandemic and must be extensively employed to improve the health of the elderly. However, some elderly cannot adapt well to these new digital communication tools; therefore, the cultivation of information technology literacy is an issue for the future [75].

In contrast, the importance of direct interpersonal interaction has also been reaffirmed by research reports that show that face-to-face interactions are more beneficial to the mental health of the elderly than digital interaction [76]. In the event of a disaster, such as the COVID-19 pandemic, rehabilitation professionals should collaborate with local public health professionals and policymakers to contribute to the development of elder-friendly community networks and recreational facilities from the perspective of a population approach to combat the social isolation of the elderly. They can also work with population-based approaches to assess the physical and cognitive abilities of individual community-dwelling elderly and consider activities, events, transportation, and communication methods that meet their unique needs [77]. Hitherto, promising practices by geriatric rehabilitation professionals have resulted in an improved participation in social activities, social interaction, QOL, and reduced healthcare costs. Still, these are often marred by organizational and institutional barriers [78,79]. Therefore, post-COVID-19 geriatric rehabilitation practice requires a community-centered approach that assesses the local stakeholder’s needs and collaborates with the local government and the community while utilizing existing welfare programs for the elderly [80].

Future research should be conducted to study different interventions, test their effectiveness, and develop strategies based on our results to prevent depression among community-dwelling elderly in the post-COVID-19 era based on the results of this study. For example, by comparing changes in the number of activities or extent of participation and the number of social networks of community-dwelling elderly after the COVID-19 pandemic and examining the causal relationship with depression, areas in need of specific support can be found in detail. In addition, direct support can be considered by finding participants who do not recover in the number of activities or extent of participation and the number of social networks among the community-dwelling elderly after the COVID-19 pandemic. Thus, longitudinal studies evaluating different factors affecting depression among community-dwelling elderly during and after the COVID-19 pandemic must examine the long-term effects of social isolation and physical activity restrictions and identify specific support.

### 4.5. Limitations

In this study, activity retention during and before the COVID-19 pandemic could be calculated retrospectively, but there were no pre-COVID-19 pandemic data for depression and social isolation. In other words, it is impossible to genuinely discuss whether the depression and social isolation scores in this study were affected by the COVID-19 pandemic. This cross-sectional survey included a small number of subjects recruited from a single senior citizens’ club in a specific area, and the club members may have been a highly active group to begin with, as indicated by the results of the JCT-IC. Participation in the club activities was freely chosen by the members; although there were some restrictions due to the prevalence of COVID-19, opportunities for social participation were provided. These factors may have introduced a selection bias, which limits the generalizability of this study.

Moreover, since this was a connectable survey and we had to get the survey questions approved by the senior citizens’ club officers, educational history and income were not included in the survey items at their request. However, when considering interventions for the elderly, attention should be paid to lifestyle, educational history, and income as critical social determinants of health [81,82]. Therefore, future studies should include these factors in their purview. Lastly, because we could not conduct a GLM with standardized independent variables, we could not compare the coefficients among the independent variables.

## 5. Conclusions

This study revealed that the COVID-19 pandemic led to a significant decline in the overall rate of activities and participation in the community-dwelling elderly in Japan. Notably, participation in high-physical-load and sociocultural activities, expected to be conducted outside the home and involve physical interactions, was decreased. This decline was in response to the government’s request suggesting people to stay at home and refrain from participating in activities to prevent COVID-19 infection. Furthermore, among the different activities, a low retention rate for activities involving a low physical load and a high degree of a lower number of social networks were associated with depression during the COVID-19 pandemic. Therefore, even in social situations where people are forced to refrain from activities due to infectious diseases or disasters, it is necessary to provide community support to enable people to enjoy activities that can be conducted indoors and foster interpersonal interactions.

## Figures and Tables

**Figure 1 ijerph-20-04228-f001:**
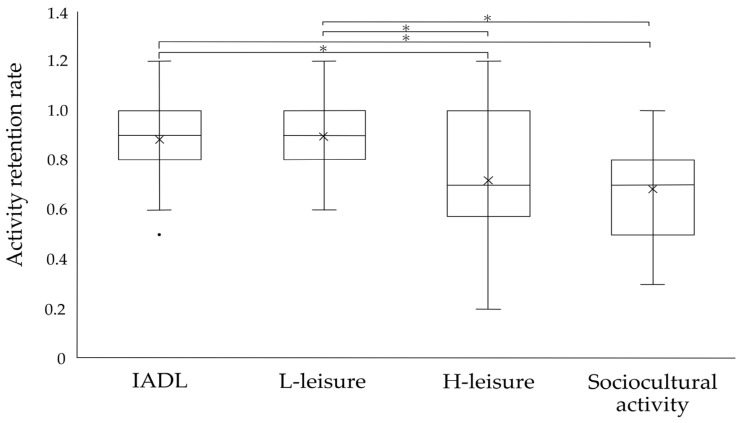
Activity retention rate during the COVID-19 pandemic. We compared the four domains measured by ACS-JPN for activity retention rate before and during the COVID-19 pandemic using Friedman’s test. Wilcoxon’s test with correction by the Bonferroni method was used for post hoc test (* *p* < 0.0083). The × indicated the mean value and the black dot indicated the outlier in the figure.

**Table 1 ijerph-20-04228-t001:** Characteristics of the participants and points of the outcome measure (n = 74).

	**n (%)**	**Mean (SD)**	**Median**	**IQR ^1^**	**Minimum**	**Max**	**Normality ^2^**
**Age (years)**	-	78.6 (5.64)	78.00	7.00	68.00	95.00	0.068
65–74	17 (23.0)	-	-	-	-	-	-
75–84	46 (62.2)	-	-	-	-	-	-
85 and above	11 (14.9)	-	-	-	-	-	-
**Sex (numbers)**							
Male	40 (54.1)	-	-	-	-	-	-
Female	34 (45.9)	-	-	-	-	-	-
**Living arrangement (numbers)**							
Living alone	20 (27.0)	-	-	-	-	-	-
Living with spouse	36 (48.6)	-	-	-	-	-	-
Living with other family units	18 (24.3)	-	-	-	-	-	-
**JST-IC (points) ^3^**		11.41 (3.17)	12.00	5.00	5.00	16.00	0.001
**GDS-15-J (points) ^4^**	-	3.14 (3.12)	2.50	4.00	0.00	12.00	0.000
**LSNS-6 (points) ^5^**	-	17.2 (5.70)	18.00	8.25	3.00	29.00	0.337
**ACS-JPN ^6^**							
***State of implementation (items)***							
IADL (26 items)	-	14.95 (4.91)	16.50	7.38	2.00	23.00	0.017
Male	-	13.60 (4.85)	13.00	8.38	5.00	22.00	0.117
Female	-	16.54 (4.56)	17.00	4.25	2.00	23.00	0.006
L-Leisure (18 items)	-	8.70 (3.17)	8.50	5.13	3.00	16.00	0.088
H-Leisure (10 items)	-	3.70 (1.87)	3.50	2.50	0.00	9.00	0.207
Sociocultural activity (18 items)	-	7.29 (3.00)	7.00	3.75	1.50	15.00	0.292
***Retention rate***							
IADL	-	0.86 (0.24)	0.90	0.20	0.50	1.20	0.000
L-Leisure	-	0.82 (0.27)	0.90	0.20	0.50	1.20	0.000
H-Leisure	-	0.67 (0.32)	0.70	0.43	0.20	1.20	0.002
Sociocultural activity	-	0.64 (0.34)	0.70	0.30	0.30	1.00	0.000

^1^ Interquartile range. ^2^ Shapiro–Wilk test *p* > 0.05. ^3^ JST-IC: Japan Science and Technology Agency Index of Competence. ^4^ LSNS-6: Lubben Social Network Scale-Japanese short version. ^5^ GDS-15-J: 15-item Geriatric Depression Scale-Japanese version. ^6^ ACS-JPN: Activity Card Sort-Japan version.

**Table 2 ijerph-20-04228-t002:** Analysis of differences in characteristics.

	Age		Sex			Living Arrangement		
	** *r_s_* **	** *p* ** **-Value**	**F**	**Z**	** *p* ** **-Value**	**F**	**H**	** *p* ** **-Value**
**GDS-15**	−0.10	0.390 ^2^	-	−0.81	0.417 ^4^	-	1.31	0.519 ^6^
**LSNS-6**	0.13	0.264 ^1^	4.11	-	0.050 ^3^	0.68	-	0.509 ^5^
**ACS-JPN**								
***State of implementation***								
IADL	−0.02	0.901 ^2^	-	−2.52	0.012 ^4^*	-	2.77	0.250 ^6^
L-Leisure	−0.14	0.220 ^1^	1.73	-	0.193 ^3^	2.33	-	0.105 ^5^
H-Leisure	−0.09	0.447 ^1^	0.09	-	0.767 ^3^	2.06	-	0.136 ^5^
Sociocultural activity	−0.09	0.443 ^1^	0.04	-	0.839 ^3^	0.60	-	0.555 ^5^
***Retention rate***								
IADL	0.07	0.580 ^2^	-	−0.06	0.950 ^4^	-	1.58	0.453 ^6^
L-Leisure	−0.04	0.764 ^2^	-	−0.41	0.683 ^4^	-	0.73	0.694 ^6^
H-Leisure	0.10	0.384 ^2^	-	−0.78	0.435 ^4^	-	5.32	0.070 ^6^
Sociocultural activity	0.12	0.292 ^2^	-	−1.14	0.253 ^4^	-	2.86	0.239 ^6^

^1^ Pearson’s Chi-square test. ^2^ Spearman’s Chi-square test. ^3^ Non-paired *t*-test. ^4^ Mann–Whitney test. ^5^ One-way analysis of variance. ^6^ Kruskal–Wallis test. (* *p* < 0.05).

**Table 3 ijerph-20-04228-t003:** Results of the generalized linear model for impact on depression (n = 74).

	Crude Model					Model 1				
	B	95% CI		Wald *χ*^2^	*p*-Value	B	95% CI		Wald *χ*^2^	*p*-Value
**ACS-JPN retention rate**										
IADL	−0.43	−1.48	0.61	0.66	0.42	−0.39	−1.43	0.65	0.53	0.47
L-Leisure	−1.41	−2.33	−0.49	9.08	0.00 *	−1.60	−2.56	−0.64	10.57	0.00 *
H-Leisure	−0.85	−0.79	0.63	0.06	0.82	−0.07	−0.77	0.64	0.03	0.86
Sociocultural activity	−0.46	−1.22	0.29	1.44	0.23	−0.38	−1.15	0.39	0.94	0.33
**LSNS-6**	−0.05	−0.07	−0.02	16.17	0.00 *	−0.05	−0.07	−0.02	15.42	0.00 *
**Age**	-	-	-	-	-	−0.02	−0.04	0.01	1.62	0.20
**Sex**	-	-	-	-	-	-	-	-	-	-
**Living arrangement**	-	-	-	-	-	-	-	-	-	-
**AIC**	365.41					365.77				
**BIC**	379.07					381.71				
	**Model 2**					**Model 3**				
	**B**	**95% CI**		**Wald** ** *χ* ** ** ^2^ **	** *p* ** **-Value**	**B**	**95% CI**		**Wald** ** *χ* ** ** ^2^ **	** *p* ** **-Value**
**ACS-JPN retention rate**										
IADL	−0.40	−1.44	0.64	0.56	0.45	−0.48	−1.57	0.61	0.73	0.39
L-Leisure	−1.60	−2.56	−0.64	10.57	0.00 *	−1.60	−2.57	−0.63	10.50	0.00 *
H-Leisure	−0.05	−0.77	0.66	0.02	0.88	−0.01	−0.75	0.73	0.00	0.98
Sociocultural activity	−0.40	−1.20	0.39	1.00	0.32	−0.41	−1.20	0.38	1.02	0.31
**LSNS-6**	−0.05	−0.07	−0.02	14.34	0.00 *	−0.05	−0.07	−0.02	14.49	0.00 *
**Age**	−0.02	−0.04	0.01	1.41	0.24	−0.02	−0.04	0.01	1.43	0.23
**Sex**	0.04	−0.25	0.32	0.07	0.80	0.02	−0.27	0.32	0.02	0.89
**Living arrangement**	-	-	-	-	-	−0.08	−0.43	0.26	0.22	0.64
**AIC**	367.71					369.49				
**BIC**	385.92					389.98				

The crude model revealed the ways in which the focal factors of this study (IADL, L-Leisure, H-Leisure, sociocultural activity, and LSNS-6) affected the participant’s depression levels without covariates of other demographical factors. Model 1 was adjusted for age, Model 2 was adjusted for age and sex, and Model 3 was adjusted for age, sex, and living arrangement. (* *p* < 0.001).

## Data Availability

The data supporting the findings of this study are available from the corresponding author, M.T. (Miki Tanikaga), upon reasonable request.

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
