# Peer review of "Changes in Community-Dwelling Elderly’s Activity and Participation Affecting Depression during COVID-19 Pandemic: A Cross-Sectional Study"

_ijerph, 2023, doi:10.3390/ijerph20054228_

Round 1

Reviewer 1 Report

Comments to the Author

I have reviewed your manuscript. Overall, the manuscript is very well-written; but there are some comments and critical questions that should be answered before publishing the paper. Please find below helpful feedback on the manuscript.

a. It seems a lot of unfamiliar abbreviation in the abstract, though you explained in the manuscript body. I suggest you rewrite simply.   

b.  We agree that the elderly groups were deprived of their outdoor activity (line 38) as well as having depression (line 46). However, I think, the authors should give some evidence to this statement. Is it caused by covid fear among community that limits outdoor activity atmosphere or due to elderly fear to do their activities. I think you can write some sentence in discussion addressing this issues. c. I am happy that you present result on the table of graph format following the MDPI template. However, Some information are missing in the figure such as what is crude model or model 1. You can explain this in method section, and why you choose this model. d. I don’t understand your purpose to present the table on appendix A. You mention retention rate of activity compared to pre covid19. I don’t see this important information on the table such as pre covid legend. How you calculate this? e. Finally. I see several color font on the references. Do you want some information to tell to the readers regarding those?

Reviewer 2 Report

The topic of the evaluated paper is important considering the pandemic restrictions that made the elderly persons isolated and inactive, which must have had an huge impact on emotional and psychic disorders, including fears and depression. The investigatiosn were performed in the initil tiume of COVID-10 pandemic (August to October 2020); the research tools used by the authors  (GDS-15, LSNS-6, and 23 ACS-JPN) seem to be adequate. In conclusions, the authors indicate the importance of maintaining the low-physical demand leisure activity and social networks at home  to prevent depression in community-dwelling elderly when they could not perform outdoor activities and keep direct interpersonal interaction.

Minor remarks: It is a pity that the authors confined the examined group to only 107 elderly persons. Even more, the real number of participants was  74 (69.2%) persons taken in the final analysis. To be quite truthful, just this number should be given in the Abstract instead of misleading number of 107 persons.

The text contained in the lines 99-113, included in the Introduction, should be rather displaced to the Discussion.

The impact of the observations would be stronger witgh the larger study population. I suggest to continue the study (maybe retrospectively)to enhance a significance of the research.
